# In Vitro Antioxidant Properties and Phenolic Profile of Acid Aqueous Ethanol Extracts from *Torreya grandis* Seed Coat

**DOI:** 10.3390/molecules27175560

**Published:** 2022-08-29

**Authors:** Wei Quan, Yang Xu, Yiting Xie, Fei Peng, Yong Lin

**Affiliations:** 1College of Food Science and Technology, Hunan Agricultural University, Changsha 410128, China; 2Department of Food Science and Technology, Hunan Food and Drug Vocational College, Changsha 410128, China; 3Engineering Technology for Utilization of Functional Ingredients from Botanicals, Hunan Agricultural University, Changsha 410128, China

**Keywords:** *Torreya grandis*, phenolic compounds, ethanol extracts, seed coat, antioxidant capacity

## Abstract

*Torreya grandis* is an important economic forestry product in China, whose seeds are often consumed as edible nuts, or used as raw materials for oil processing. To date, as an important by-product of *Torreya grandis*, comprehensive studies regarding the *Torreya grandis* seed coat phenolic composition are lacking, which greatly limits its in-depth use. Therefore, in the present study, the *Torreya grandis* seed coat was extracted by acid aqueous ethanol (TE), and NMR and UHPLC-MS were used to identify the major phenolics. Together with the already known phenolics including protocatechuic acid, catechin, epigallocatechin gallate, and epicatechin gallate, the unreported new compound 2-hydroxy-2-(4-hydroxyphenylethyl) malonic acid was discovered. The results of the antioxidant properties showed that both TE and 2-hydroxy-2-(4-hydroxyphenylethyl) malonic acid exhibited strong ABTS, DPPH, and hydroxyl radical-scavenging activity, and significantly improved the O/W emulsion’s oxidation stability. These results indicate that the TE and 2-hydroxy-2-(4-hydroxyphenylethyl) malonic acid could possibly be used in the future to manufacture functional foods or bioactive ingredients. Moreover, further studies are also needed to evaluate the biological activity of TE and 2-hydroxy-2-(4-hydroxyphenylethyl) malonic acid to increase the added value of *Torreya grandis* by-products.

## 1. Introduction

*Torreya grandis* (*Taxaceae*), also known as Chinese *Torreya*, is a large evergreen tree native cultivated in China [1,2]. Currently, *Torreya grandis* is an important economic tree species in China, and the cultivation area of *Torreya grandis* in China has reached 520,000 hectares [3,4]. An earlier study revealed that this plant contains a high level of unsaturated fatty acids including oleic and linoleic acids [5,6,7]. Due to its high nutritional value and economic value, *Torreya grandis* is a popular choice for people who are looking for healthy and nutritious food [1]. Furthermore, in our previous study, it was revealed that the seeds of *Torreya grandis* contained various nutrients, proteins, and a variety of bioactive components such as sterols, polyphenols, and tocopherols [6,8].

In addition to being widely consumed as edible nuts, *Torreya grandis* has also been considered as a highly nutritious and health-promoting edible oil source in China [5,9,10]. Therefore, previous studies have mainly focused on the extraction, processing, and utilization of oil from *Torreya grandis* seeds, which leads to the inevitable production of many by-products such as the aril, seed coat, and meal during the processing of *Torreya grandis* [7,11,12].

From the chemical composition of *Torreya grandis*, these by-products can be processed as potential sources of valuable phenolic compounds and proteins, which exhibited a variety of biological properties including antioxidant, anti-inflammatory, anti-atherosclerotic, etc. [8,13,14,15]. Thus, previous studies have investigated the utilization of by-products of *Torreya grandis*. Among them, essential oil from *Torreya grandis* aril and their odor properties and volatile compounds have been fully investigated [12,16,17]. Furthermore, researchers have also focused on and reported the functional properties of *Torreya grandis* aril essential oil such as antioxidant, antitumor effect, etc. [14,18]. Furthermore, other studies have also screened the bioactive peptide from *Torreya grandis* meal protein hydrolysates [11]. Our previous study prepared the enzymatic hydrolysates from *Torreya grandis* meal protein and investigated the antioxidant properties of those enzymatic hydrolysates [19]. However, to the best of our knowledge, there is limited research focused on the composition, functional properties, and in-depth processing and utilization of the *Torreya grandis* seed coat.

According to our preliminary study results, *Torreya grandis* seed coats are rich in phenolic compounds (about 530.6 mg/g), which indicates a potential value for in-depth processing and utilization. Therefore, the present study aimed to prepare a *Torreya grandis* coat extract, identify the main phenolic compounds in it, and analyze its antioxidant properties, for the better utilization of by-products of *Torreya grandis*.

## 2. Results and Discussion

### 2.1. Identification of Phenolic Compounds of Ethanol Extract from Torreya grandis Seed Coat

The results (Appendix A) showed that the total phenolic and flavonoid content of the ethanol extract from the *Torreya grandis* seed coat (TE) was 530.6 mg GA/g and 25.5 mg RE/g, respectively. Considering that the TE contains abundant phenolic compounds, the present study further identified the main phenolic compounds in the TE by UHPLC-PDA-QTOF-MS/MS. The identification of the phenolics was mainly achieved by first comparing their retention time and the MS data with those reported in the current references, and further confirming the results using standard chemicals for phenolics.

In the UHPLC-PDA-QTOF-MS/MS chromatogram (Appendix A) of the ethanol extract from the *Torreya grandis* seed coat, several peaks (peaks F1, F3, F4, and F5) were readily identified as protocatechuic acid, catechin, epigallocatechin gallate, and epicatechin gallate, respectively. The detailed profiles of the identified phenolics including the retention times, fragment ions, molecular formula, compound name, CAS number, and chemical structure formula are presented in Table 1.

However, unlike the other above reported phenolics, a chromatographic peak (Appendix A) was judged to be a new compound in *Torreya grandis* or a completely novel compound discovered for the first time with its mass spectra showing quasimolecular ions at m/z 240.1470 in the positive ion mode (Table 1), thus C_11_H_12_O_6_ was suggested as its molecular formula. Therefore, in order to determine the chemical structure of F2, we further performed an in-depth NMR-based analysis.

First, the number of carbons in the F2 compound was further confirmed by ^1^H–^13^C HMBC 2D NMR (Figure 1, Appendix A). A ^1^H–^1^H coupling system in the ^1^H–^1^H COSY spectrum suggested the presence of an aromatic one (as shown in Appendix A). In addition, as shown in Figure 1A, the ^1^H NMR of compound F2 showed two typical ABX coupling systems with four proton signals δ_H_ 7.087 (2H, dd, *J* = 8.8 Hz), δ_H_ 6.709 (2H, dd, *J* = 8.4 Hz), δ_H_ 2.998 (2H, *J* = 14 Hz, H-4), and δ_H_ 2.593 (2H, *J* = 16 Hz, H-3), which also confirmed the existence of the –CH2–CH2- structure in this compound. Moreover, in accordance with the molecular formula of compound F2, in the ^13^C NMR spectrum, nine carbons including two carbonyls: δ178.35 (C = O, C-1), δ174.84 (C = O, C-1′); 4 sp2 carbons which involved in aromatic ring: δ157.92 (C, C-8), δ133.15 (CH, C-10), δ128.24 (C, C-5), δ116.4 (CH, C-9); one methine: δ77.47 (C, C-2), and two methylene: δ46.03 (CH2, C-4), δ44.27 (CH2, C-3) were detected (Figure 1B). Compared with the molecular formula C_11_H_12_O_6_, it was found that the two carbon signals were missing, indicating that these two carbons had carbons with the same chemical environment. An analysis of the ps-HSQC spectrum showed that each proton was associated with the respective carbon in Appendix A. Overall, the structure of F2 was determined as 2-hydroxy-2-(4-hydroxyphenylethyl) malonic acid (Figure 1C).

### 2.2. Antioxidant Activity of Ethanol Extract from Torreya grandis Seed Coat

Although *Torreya grandis* is known for its antioxidant capacity [20], limited information has been reported on the antioxidant activity of the *Torreya grandis* seed coat. Therefore, the antioxidant activity of TE and the newly identified compound F2 (2-hydroxy-2-[4-hydroxyphenylethyl] malonic acid) was evaluated by five different assays.

As shown in Figure 2, the antioxidant activity of TE and F2 increased with the increasing concentration of TE and F2. In detail, the ABTS scavenging ability of TE was similar to that of the control, and the free radical scavenging rate of TE could reach 93.4%, even when the sample concentration was 0.5 mg/mL, which might be attributed to the abundant phenolic compound content in the TE (Figure 2A). However, F2 exhibited a relatively low ABTS free radical scavenging activity at lower concentrations, and a 91.3% ABTS free radical scavenging rate was only achieved when the concentration reached 5 mg/mL. The DPPH free radical is an oil-soluble free radical, which mainly accepts electrons or hydrogen from antioxidants and becomes a stable product [21]. As Figure 2B shows, at the tested concentrations, TE and the control exhibited comparable DPPH scavenging capacity at the same concentrations (*p* > 0.05). When the concentration of F2 increased to 0.5 mg/mL, its DPPH radical scavenging rate could also reach 96.5%, which was a similar level to TE and the control. Moreover, various diseases are caused by oxidative stress induced by hydroxyl radicals, which are generated from the Fenton reaction, so scavenging them is imperative in protecting against them [22]. According to Figure 2C, TE exhibited a significantly higher (*p* < 0.05) hydroxyl radical scavenging activity than that of the control when the added concentration of TE was higher than 0.05 mg/mL. Similar to the results of the ABTS free radical scavenging ability, the hydroxyl radical scavenging rate of F2 was only 20% when its addition amount was 0.05–1 mg/mL, which was significantly lower than that of TE and the control. When the added concentration of F2 was greater than 1 mg/mL, the hydroxyl radical scavenging rate increased rapidly, and finally reached more than 90%.

The metal chelating ability mainly reflects a compound that stabilizes oxidized forms of metal ions by reducing their redox potential, which is also a valuable indicator of antioxidant activity [23]. As shown in Figure 2D, the metal chelating ability of TE and F2 at the concentration of 0.01–5 mg/mL exhibited a dose dependent effect, and F2 even exhibited significantly higher metal chelation ability than TE. However, the metal chelating ability of TE and F2 was significantly lower than that of the control at the tested concentrations. Moreover, the reducing power of TE at the concentration of 0.01–5 mg/mL also exhibited a dose dependent effect (Figure 2E). As the TE concentration increased above 1 mg/mL, its reducing power became significantly higher (*p* < 0.05) than the control. However, the reducing power of F2 was significantly lower than that of the control and TE at the tested concentration conditions.

In conclusion, TE was significantly more effective in scavenging the ABTS, DPPH, and hydroxyl radicals, but exhibited relatively weak reducing power and metal chelating activity. F2 also showed potential antioxidant activity, especially when it was added at a concentration greater than 1 mg/mL.

### 2.3. Effect of Ethanol Extract from Torreya grandis Seed Coat on Lipid Oxidition in an O/W Emulsion

In our preliminary experimental results, the ethanol extract from the *Torreya grandis* seed coat exhibited a strong antioxidant capacity. Therefore, TE and F2 was added into an O/W emulsion, in which A1 and A2 represent the addition of 0.1 and 0.2 mg/mL TE, respectively; B1 and B2 represent the addition of 0.1 and 0.2 mg/mL in the newly identified compound 2-hydroxy-2-(4-hydroxyphenylethyl) malonic acid (F2), respectively. Finally, the pH value and TBARS value of the emulsion were determined.

As shown in Figure 3A, the initial pH of all samples was 7.12, after 23 days of storage, the pH value of the control group was significantly decreased from 7.12 to 4.73. Compared to the control, the pH value of the emulsion added with TE at the concentration of 0.1 mg/mL and 0.2 mg/mL exhibited a non-significant decrease during storage, which was always maintained at pH 7.0. In addition, the particle size of the emulsion added with TE and F2 was 250–270 nm during storage, significantly lower than that of the control (Figure 3B).

In order to evaluate the emulsion’s lipid oxidative stability, the TBARS value was used as previous reported [24]. As shown in Table 2, compared with the control group, with the increase in the storage days, the emulsion added with TE and F2 showed a stronger lipid oxidative stability. The O/W emulsion TBARS production was lower than 0.42 mg/kg during the first three days of storage. However, the TBARS value increased significantly after storage for 10 days and reached 13.1 and 20.7 mg/kg after storage for 18 and 23 days, respectively, which was due to the increasing production of fat oxidation secondary products. Moreover, the increase in the aldehydes, ketones, and organic acid compounds in the O/W emulsion has also prompted a significant decrease in the pH of the system, which is consistent with the result that the pH of the control group was significantly decreased. With the addition of TE and F2, a significantly lower level of TBARS was produced by the O/W emulsions than in the control groups, and the TBARS value significantly decreased by 94.9% and 92.1%, respectively, after storage for 23 days. 

## 3. Conclusions

In order to better develop and utilize the by-products produced in the processing of *Torreya grandis,* in the present study, the seed coat of commercially available *Torreya grandis* was extracted by acidic aqueous ethanol, its phenolic profile was characterized, and the antioxidant activity analyzed. Five major phenolics were isolated and identified by means of UHPLC-MS and NMR experiments. Four turned out to be already known molecules. The remaining unreported compound was identified as 2-hydroxy-2-(4-hydroxyphenylethyl) malonic acid. Both TE and 2-hydroxy-2-(4-hydroxyphenylethyl) malonic acid exhibited varying abilities in scavenging ABTS, DPPH+, and hydroxyl radicals, and effectively improved the oxidation stability of the O/W emulsions. More in-depth studies on the bioactivities of TE and 2-hydroxy-2-(4-hydroxyphenylethyl) malonic acid need to be conducted in the future, since the occurrence of such molecules might amplify the nutritive value of *Torreya grandis*.

## 4. Materials and Method

### 4.1. Materials

The *Torreya grandis* (Taxaceae) seed coat is a by-product of *Torreya grandis* seeds and was provided by a local supplier (Zhejiang Province, China). DPPH, ABTS, potassium ferricyanide, ferrous sulfate, ferrous chloride, 1,10-phenanthroline, ferrozine, and other reagents were purchased from Merck (St. Louis, MO, USA).

### 4.2. Preparation of Ethanol Extracts from Torreya grandis Seed Coat

At first, the *Torreya grandis* seed coat was ground into powder using liquid nitrogen, and the ground powder was passed through a 120-mesh sieve to obtain the *Torreya grandis* seed coat samples. Then, refering to some previous studies with slight modification [25], the sample was extracted with 1% acetic acid–ethanol solution (ethanol:water:acetic acid, 70:29:1, *v*/*v*/*v*) at a ratio of 1:20 (*w*/*v*) by stirring and ultra-sonication (40 Hz) assisted for 2 h at 30 °C, and the mixture was centrifugated at 11,000 g for 12 min. The filtered residue was repeatedly extracted three times, and the supernatants were collected and combined. By vacuum rotary evaporation at 45 °C, ethanol from the extracts was removed, water was added to the final concentration, and a 0.45 mm membrane was used to filter the solution. The filtrate was vacuum freeze-dried to obtain the *Torreya grandis* seed coat extract, which was defined as TE and used in all of the subsequent analyses.

### 4.3. Isolation and Identification of Phenolic Compounds of Ethanol Extracts from Torreya grandis Seed Coat

#### 4.3.1. Isolation of Phenolic Compounds Using Preparative High-Performance Liquid Chromatography

The TE extract (500 μL) was fractionated on a Waters 2545 HPLC system and 2489 UV detector, an autosampler, and a Waters 2767 auto-fraction collector using a TSK-gel ODS C18 column (250 × 5 mm i.d., 5 μm) with the following program of gradient elution: 0 min, 5% B; 15 min, 40% B; 20 min, 80% B; 22 min, 100% B; 23 min, 5% B; 30 min, 5% B, in which mobile phase A is 0.1% formic acid (dissolved in ultra-pure water) and mobile phase B is acetonitrile, the flow rate was set to 3.0 mL/min, and the detection wavelength was 260 to 280 nm. A total of 20 fractions were collected, and equal fractions were pooled and dried in a vacuum concentrator for further analysis.

#### 4.3.2. Identification of Phenolic Compounds Using UPLC-PDA-QTOF-MS

All fractions collected from preparative HPLC were first separated using a Waters UPLC system equipped with a Waters BEH C18 (2.1 mm × 100 mm, 1.7 µm) column and 2996 DAD detector, and the gradient elution was composed of mobile phases A (0.1% formic acid–water) and B (acetonitrile), and the program was set as follows: 0–2 min, 2% A, 2–20 min, 2–30% A, 20–24 min, 30–80% A, 24–26 min, 80–100% A, 26–30 min, 100–2% A, flow rate, and injection volume were set to 1.0 mL/min and 10 µL, respectively. Column temperature and the detector wavelengths were 35°C and 280 nm, respectively. Then, the samples were analyzed on a Waters Quattro Premier mass spectrometer (Milford, MA, USA) and the instrument parameters was set a follows: source temperature (110 °C), desolvation temperature (400 °C), capillary voltage (3.5 kV), cone voltage (30 V), the mass range scanned (20–1500 m/z).

#### 4.3.3. Identification of Unknown Compounds with NMR

Each combined fraction was solubilized in 500 μL of methanol-d4 and transferred into a 5 mm NMR tube for 1 H NMR analysis. Spectra were collected at 300 K on an Bruker Avance 800 MHz spectrometer (Coventry, UK) using a 5 mm cryogenic TCI probe. ^1^H NMR spectra were recorded with scans: 16–32; data point: 64 k; spectral width: 12 ppm, and recycle delay: 3.28. 2D NMR spectra including ^1^H–^1^H COSY, TOCSY, and ^1^H–^13^C HSQC were recorded according to previously reported studies with slight modification: 2048 data points and four scans for each of the 256 increments. For ^1^H–^13^C HMBC 2D NMR spectra, 2048 data points and 16–48 scans for each of the 512 increments were adopted. In addition, ^1^JC–H and ^n^JC–H were set to 145 Hz and 6 Hz for both the HSQC and HMBC experiments. The NMR spectra were processed with MestRenova 9.0 software, the phase and baseline were manually corrected, and the chemical shifts were referenced to the solvent signals of methanol-d4 (δ_H_ = 3.31; δ_C_ = 49.0 ppm).

### 4.4. Antioxidant Activity Assays

The TE powder or ascorbic acid (control) was dissolved in deionized water with a concentration series of 0.05, 0.10, 0.50, 1.0, and 5.0 mg/mL, respectively. The antioxidant activities of the TE were measured by five different free radical scavenging activity assays including ABTS, DPPH, hydroxyl assays, metal chelating activity, and reducing powder according to previous reports with slight modification. The results are expressed as the free radical inhibition rate (%) or metal ion chelation rate (%) [19,26,27,28,29].

As reported in our previous studies [27,29], at first, the ABTS radical cation solution was obtained by 7 mM ABTS solution with potassium persulfate (2.45 mM), which reacted in the dark at 4 °C for 15 h. Then, the ABTS working solution was prepared by the ABTS radical cation solution, which was diluted with 5 mM PBS (pH 7.4) to an absorbance of 0.70 ± 0.02 at 734 nm. Finally, each concentration of TE (0.5 mL) was reacted with 9.5 mL of ABTS working solution in the dark for 10 min, and the absorbance of the reacted solution was measured at 734 nm.

The scavenging activity against the DPPH free radicals was performed according to the previously reported method [19,29]: Each concentration of TE (0.5 mL) was mixed with 500 µL of the 0.1 mM DPPH radical solution (dissolved in 70% methanol) and incubated for 30 min in the dark at 30 °C, then the absorbance of the mixtures was recorded at 517 nm.

According to Qie et al. and Wang et al. [26,28], the hydroxyl radical scavenging activity was measured as per the following procedure. Each concentration of TE (1 mL) was sequentially mixed with 0.5 mL 0.1 M sodium phosphate buffer (pH 7.4), 5 mM 1,10-phenanthroline, 5 mM FeSO_4_, and 0.03% H_2_O_2_, and then reacted at 35 °C for 1 h; the absorbance of the reacted mixture solution was measured at 536 nm.

We refer to our previously mentioned method [19,27]: 1 mL TE sample, 0.05 mL 2 mM FeCl_2_, 1.85 mL distilled water, and 0.1 mL 5 mM ferrozine were mixed together and reacted at 30°C for 1 min, then the absorbance of the reacted mixture solution was measured at 562 nm to calculate the activity of the metal chelating activity.

Based on our previous study, we estimated the reducing power as follows [19]: 0.5 mL of the TE sample was sequentially mixed with 0.5 mL of 0.2 M, pH 6.6 phosphate buffer, and 0.5 mL of 1.0% K_3_[Fe(CN)_6_] solution and reacted in a water bath at 50 °C for 20 min. To stop the reaction, 500 μL of 10% trichloroacetic acid was added, and the reacted solution was centrifuged at 1500× *g* for 10 min. A supernatant was mixed with distilled water and 0.1% FeCl_3_, and the absorbance at 700 nm was measured after 10 min of reaction at 25 °C.

### 4.5. Antioxidant Activity of Ethanol Extract from Torreya grandis Seed Coat Applied in Oil in Water (O/W) Emulsion

#### 4.5.1. Emulsion Preparation

Following previous methods, to obtain O/W emulsions with no oxidizable fatty acids, 1.0% soybean oil was dissolved in a sodium acetate/imidazole buffer solution (pH 7.0) [19], in which Tween 20 was added at a 1:10 emulsifier/oil ratio as an emulsifier. After adding 0, 1, or 5 mg/mL TE to the above mixed solution, the mixed solution was sheared and dispersed at 21,500 rpm using an Ika T18 disperser (Staufen, Germany). The mixed solution was then homogenized twice at 36 MPa using a high pressure homogenizer to prepare O/W emulsions, and to prevent microbial growth, all emulsions were treated with sodium azide at 3 mM. Finally, the prepared emulsion was put into a glass bottle with a screw cap and stored in a 37 °C incubator away from the light. The above emulsion samples with the addition of 0, 1, and 5 mg/mL TE were defined as the control TE1 and TE2, respectively.

#### 4.5.2. Measurement of the Particle Size of Emulsion

Referring to the method reported in our previous study, all emulsion samples in this study were first diluted 50-fold using 10 mM sodium acetate/imidazole buffer solution (pH 7.0), and then transferred to a 3 mL plastic cuvette to determine the particle size of the emulsion by Malvern ZEN3600 dynamic light scattering (Worchester, UK), where the results of the emulsion particle size are expressed as the Z-average mean diameter.

#### 4.5.3. Measurement of Emulsion Oxidative Stability

Based on previous studies [19,24,30], the oxidative stability of the emulsion was evaluated by measuring the lipid peroxidation inhibition capacity, which was revealed as the TBARS value of emulsion. A total of 1 mL of the sample was reacted with a mixed solution that contained 15% TCA, 0.37% TBARS, and 1.8% HCl at 100 °C for 30 min. As a next step, the reacted solution was cooled rapidly in an ice bath, centrifuged at 3000× *g* r/min for 10 min, and the absorbance of the supernatant was measured at 534 nm; the results were expressed TBARS mg/kg.

### 4.6. Statistical Analysis

All experiments were performed in triplicate, and results are reported as the mean ± SD. The least significant difference test was used to determine whether there was a significant difference (*p* < 0.05).

## Figures and Tables

**Figure 1 molecules-27-05560-f001:**
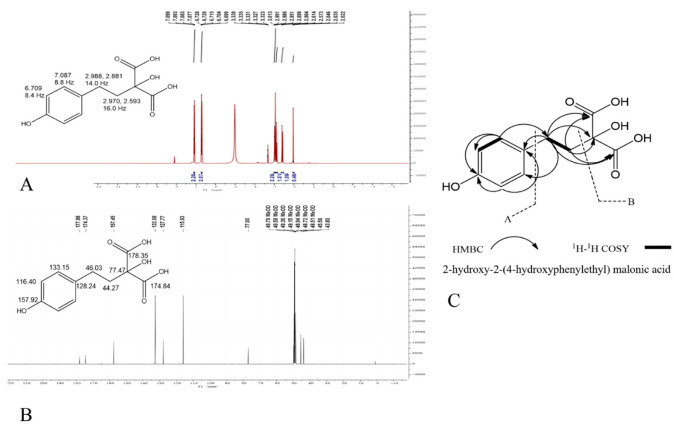
The identification of peak F2 based on (**A**) ^1^H NMR of F2, (**B**) ^13^C NMR of F2. (**C**) The key HMBC and 1H-1H COSY correlation of compound F2.

**Figure 2 molecules-27-05560-f002:**
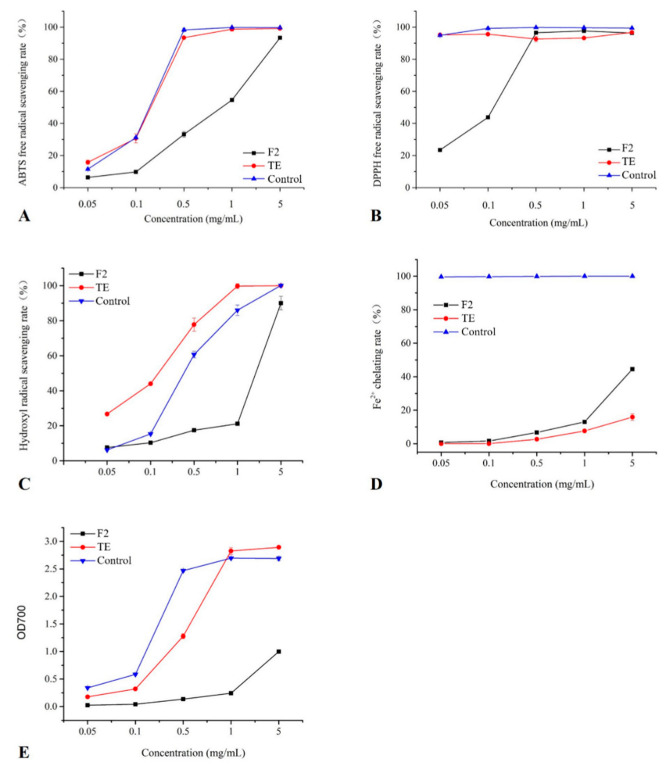
The antioxidant scavenging capacity of the ethanolic extract and newly identified compounds from the seed coat of *Torreya grandis* as determined by (**A**) ABTS scavenging activities, (**B**) DPPH· scavenging activities, (**C**) hydroxyl radical scavenging activities, (**D**) Fe^2+^ chelating activity, and (**E**) reducing power.

**Figure 3 molecules-27-05560-f003:**
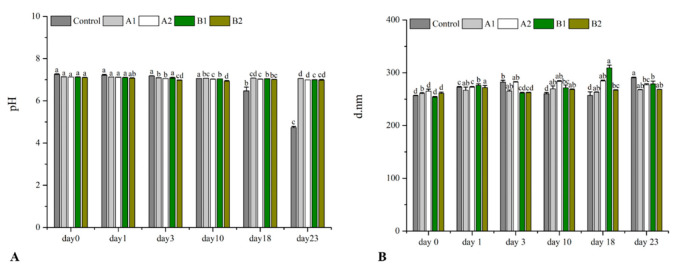
The effect on the O/W emulsion particle size and pH at different concentrations of ethanolic extract and newly identified compounds from the seed coat of *Torreya grandis*. (**A**) the initial pH, (**B**) the particle size of the emulsion, A1: 0.1 mg/mL TE; A2: 0.2 mg/mL TE; B1: 0.1 mg/mL F2; B1: 0.2 mg/mL F2. The same additive and the same concentration with different letters indicate a significant difference (*p* < 0.05).

**Table 1 molecules-27-05560-t001:** The identification of phenolic compounds from the ethanolic extract of the *Torreya grandis* seed coat by UPLC-QTOF-MS.

Peak	R_t_	M^+^	Fragment Ions	Molecular Formula	Compounds	CAS	Structural Formula
	(min)	(*m/z*)	(*m/z*)				
1	9.52	154.1	109.1, 69.0	C_7_H_6_O_4_	Protocatechuic acid	99-50-3	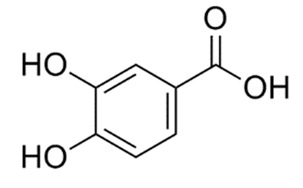
2	10.80	240.1	179.0, 149.1, 107.0	C_11_H_12_O_6_	2-hydroxy-2-(4-hydroxyphenylethyl) malonic acid	-	-
3	11.15	290.2	245.2, 205.1, 125.1	C_15_H_14_O_6_	Catechin	154-23-4	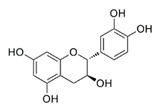
4	11.94	458.3	305.2, 287.2, 169.1, 125.1	C_22_H_18_O_11_	Epigallocatechin gallate	989-51-5	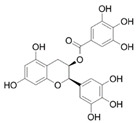
5	12.57	442.3	289.2, 169.1	C_22_H_18_O_10_	Epicatechin gallate	1257-08-5	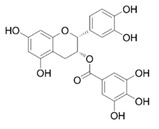

**Table 2 molecules-27-05560-t002:** The effect of the ethanolic extract and newly identified compounds from the seed coat of *Torreya grandis* on the content of MDA in the O/W emulsion (TBARS mg/kg sample).

	Concentration (mg/mL)	Storage Time (Day)					
		Day 0	Day 1	Day 3	Day 10	Day 18	Day 23
TE	0	0.22 ± 0.02 ^Ad^	0.22 ± 0.01 ^ABd^	0.42 ± 0.05 ^Ad^	1.34 ± 0.07 ^Ac^	12.80 ± 0.53 ^Ab^	20.44 ± 0.09 ^Aa^
	0.1	0.19 ± 0.01 ^Ad^	0.19 ± 0.01 ^Bd^	0.20 ± 0.03 ^Bd^	0.53 ± 0.12 ^Bc^	0.82 ± 0.05 ^Bb^	1.04 ± 0.06 ^Ba^
	0.2	0.22 ± 0.03 ^Ad^	0.30 ± 0.04 ^Acd^	0.18 ± 0.01 ^Bd^	0.42 ± 0.14 ^Bc^	0.74 ± 0.04 ^Bb^	0.93 ± 0.05 ^Ba^
F2	0	0.22 ± 0.02 ^Ad^	0.22 ± 0.01 ^Ad^	0.39 ± 0.05 ^Ad^	1.34 ± 0.07 ^Ac^	12.80 ± 0.53 ^Ab^	20.44 ± 0.09 ^Aa^
	0.1	0.10 ± 0.03 ^Bb^	0.11 ± 0.02 ^Bb^	0.29 ± 0.04 ^Bb^	0.70 ± 0.18 ^Bb^	1.27 ± 0.61 ^Ba^	1.61 ± 0.27 ^Ba^
	0.2	0.14 ± 0.02 ^ABc^	0.19 ± 0.03 ^Ac^	0.22 ± 0.06 ^Cc^	0.50 ± 0.11 ^Bb^	0.97 ± 0.28 ^Bab^	1.17 ± 0.10 ^Ba^

Different lowercase letters indicate significant differences between results in the same row, while different capital letters indicate significant differences between the results in the same column (*p* < 0.05).

## Data Availability

The data presented in this study are available in Appendix A.

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
