# Peer review of "In Vitro Antioxidant Properties and Phenolic Profile of Acid Aqueous Ethanol Extracts from Torreya grandis Seed Coat"

_molecules, 2022, doi:10.3390/molecules27175560_

Round 1
Reviewer 1 Report
There is occasional dissimilarity in writing styles, however I endorsed their publication.
Author Response
Q: There is occasional dissimilarity in writing styles, however I endorsed their publication
R: Thank you very much. I really appreciate for your reviewing work of my manuscript and giving us important comments.

Reviewer 2 Report
The manuscript "In vitro antioxidant properties and phenolic profile of acid aqueous ethanol extracts from Torreya grandis seed coat" devoted to comprehensive study of composition of seed coat of commercially available Torreya grandis. Acidic aqueous ethanol was used as the solvent. Ultrasound-assisted extraction was used to obtain extract. A lot of methods for characterization of the extract were used, such as HPLC, NMR, various methods of antioxidant activity assay etc. A new compound, 2-hydroxy-2-(4-hydroxyphenylethyl) malonic acid was found. This study will be interesting for the phytochemists and the specialists in food chemistry and Pharmacognosy.
The manuscript is written clearly, well-structured and has a good scientific soundness. This manuscript can be published in the Molecules journal after minor revision taking into account some of the remarks described below
1. Section 2.3.2: “…mobile phase B is acetonitrile (B),…”. “(B)” – is not necessary.
2. Section 2.4. Antioxidant activity assays: Check, please, if all reagents are mentioned in the Section 2.1. Materials.
3. Check throughout the text whether all abbreviations are deciphered. “TCA” should be described.
4. Section 3.2.: Incorrect numbering of the Figures. Antioxidant activity is presented in Fig. 3, not in Fig.2. Also, what is the “control” sample in the DPPH and other methods? Usually, control is the ethanol solution, and it give the inhibition of DPPH about 1-2%, but not >90%. In DPPH method values on the inhibition > 80-90% are not indicative and it is necessary to find the concentration near IC50 to compare samples (or “control”?). Ideally, the Fig. 3B should look like 3A.
5. Table 2.: I think, the letters should be in superscript.
Author Response
Q: Section 2.3.2: “…mobile phase B is acetonitrile (B),…”. “(B)” – is not necessary.
R: Thank you very much. I have revised this sentence (see Line 106 in revised manuscript).
Q: Section 2.4. Antioxidant activity assays: Check, please, if all reagents are mentioned in the Section 2.1. Materials.
R: Thank you so much. Some important reagents used in the antioxidant experiments in Section 2.4, such as ABTS, DPPH, etc., are listed in Section 2.1 Materials and Methods. But some conventional reagents such as hydrogen peroxide, ferrous chloride, etc. are not listed in 2.1 Materials and methods section. But we also stated in section 2.1 that all other analytical grade reagents were purchased from Sinopharm Chemical Reagent Co., Ltd (see Line 74-77 in revised manuscript).
Q: Check throughout the text whether all abbreviations are deciphered. “TCA” should be described.
R: Thank you for your good suggestion. I have added the full name of TCA, and checked throughout the text about the describe of abbreviations (see Line 155 in revised manuscript).
Q: Section 3.2.: Incorrect numbering of the Figures. Antioxidant activity is presented in Fig. 3, not in Fig.2. Also, what is the “control” sample in the DPPH and other methods? Usually, control is the ethanol solution, and it give the inhibition of DPPH about 1-2%, but not >90%. In DPPH method values on the inhibition > 80-90% are not indicative and it is necessary to find the concentration near IC50 to compare samples (or “control”?). Ideally, the Fig. 3B should look like 3A.
R: Thank you so much for your good suggestion. I feel sorry that the design of the control group in the antioxidant experiments was not clearly stated in the present manuscript. In order to more intuitively reflect and compare the antioxidant capacity of Torreya grandis seed coat extract, referring to some previous literature reports, we chose the same concentration of ascorbic acid as the control (see Line 126-131 in revised manuscript).
Here are the references:
- Wang, Z., Liu, X., Xie, H., Liu, Z., Rakariyatham, K., Yu, C., & Zhou, D. (2021). Antioxidant activity and functional properties of Alcalase-hydrolyzed scallop protein hydrolysate and its role in the inhibition of cytotoxicity in vitro. Food Chemistry, 344, 128566.
- Singh, G., Pathania, R., Khan, M., Tonk, R. K., Kumar, D., & Dash, A. K. (2021). Identification and quantification of some natural compounds of Pinus gerardiana leaf extract and its antimicrobial and antioxidant activities. Pharmacologyonline, 2, 333-351.
- Boulmokh, Y., Belguidoum, K., Meddour, F., & Amira-Guebailia, H. (2021). Investigation of antioxidant activity of epigallocatechin gallate and epicatechin as compared to resveratrol and ascorbic acid: Experimental and theoretical insights. Structural Chemistry, 32(5), 1907-1923.
- Liang, X., Cao, K., Li, W., Li, X., McClements, D. J., & Hu, K. (2021). Tannic acid-fortified zein-pectin nanoparticles: Stability, properties, antioxidant activity, and in vitro digestion. Food Research International, 145, 110425.
- Rajauria, G., Ravindran, R., Garcia-Vaquero, M., Rai, D. K., Sweeney, T., & O'Doherty, J. (2021). Molecular characteristics and antioxidant activity of laminarin extracted from the seaweed species Laminaria hyperborea, using hydrothermal-assisted extraction and a multi-step purification procedure. Food Hydrocolloids, 112, 106332.
Q: Table 2.: I think, the letters should be in superscript.
R: Thank you so much for your kind advice, I have revised the table 2 to make the letters in superscripted (see Line 401-404 in revised manuscript).

Reviewer 3 Report
The paper is a communication about the properties of Torreya grandis seed coat and a further emulsion. However, does not the title nor the abstract are clear about the latter. Moreover, the isolation and further use of a new compound is reported but also no reflected properly on the abstract. I strongly recommend to rewrite both sections.
Major corrections:
-It should be revise the section 2.2, it does not explain properly the extract procedure.
- In general, figures are overload with information, please consider those relevant to be included in the main text, and move some to SM.
- The chemical structures reflected on Figure 1 should be included in Table 1 for a better comprehension.
- Picture of seed (figure 1 A) could be in SM.
- Please revise important information of Figure 2. The chemical structure is presented 4 times in the figure. Moreover, the size of the figure does not allow for a better interpretation. Suggestion to move chromatogram to SM and other less relevant information.
- In Section 3,2 discusses Figure 2, but it is in fact figure 3.
- Figure 3 legend indicate an F figure but there are only E.
-Legend in figure 4 should be improved in order to better establish what means A1, A2, B1 and B2.
- Table 2 has multiple grammar error, from misspelling concentration to the use of lowercase for the statistical representation.
- Bibliographic search should be improved since only few recent papers are listed.
Minos correction
- Please use the proper italics when naming Torreya grandis seed coat.
- Correct throughout the text error such as CD3OD, ABTS+, among other .
Author Response
Q: It should be revise the section 2.2, it does not explain properly the extract procedure
R: Thank you so much for your good suggestion. I feel sorry that the section 2.2 did not explain properly the extract procedure, and I have revised this section (see Line 79-89 in revised manuscript).
Q: In general, figures are overload with information, please consider those relevant to be included in the main text, and move some to SM.
R: Thank you very much for your suggestion. According to your suggestion, I have revised the figures and the legend of figures, and moved some to supplementary materials (see Line 388-397, and 405-407 in revised manuscript and supplementary Figures in supplementary materials).
Q: The chemical structures reflected on Figure 1 should be included in Table 1 for a better comprehension.
R: Thank you very much for your comments. I completely agree with you that it would be more clear to put the structural formula of the compound into Table 1, and I have revised table 1 (see Line 398-400 in revised manuscript).
Q: Picture of seed (figure 1 A) could be in SM.
R: Thank you so much for your good suggestions. I totally agree with your suggestion as Figure 1A is not presented in the main body of the manuscript. In addition, according to your suggestion, the structural formula of the compound is put in Table 1, therefore, both Figure 1A and Figure 1B are put in the supplementary material (see Line 398-400 in revised manuscript and Supplementary Figure 1 in supplement material).
Q: Please revise important information of Figure 2. The chemical structure is presented 4 times in the figure.
Moreover, the size of the figure does not allow for a better interpretation. Suggestion to move chromatogram to SM and other less relevant information.
R: Thank you very much for your comment. The chemical structure in Figure 2 exhibited the important information from 1H NMR and 13C NMR of F2, and I have revised figure 2 to make the figure more clear. Meanwhile the raw chromatogram of UPLC-MS and NMR have been move into supplementary material (see Line 405-407 in revised manuscript and Supplementary Figure 2 to Figure 7 in supplement material).
Q: In Section 3,2 discusses Figure 2, but it is in fact figure 3.
R: Thank you very much for your comment. I have revised those sentences that showed incorrect numbering of the figures (see Line 230, 234, 238, 243, 252, and 256 in revised manuscript).
Q: Figure 3 legend indicate an F figure but there are only E.
R: Thank you very much for your comment. I feel sorry for my mistake, it should be E, and I have revised this sentence (see Line 256 in revised manuscript).
Q: Legend in figure 4 should be improved in order to better establish what means A1, A2, B1 and B2.
R: Thank you very much for your comment. I have revised the legend in Figure 4 to better establish what means A1, A2, B1 and B2, moreover, I have added some sentences in the main text to introduce the meaning of A1, A2, B1 and B2 (see Line 267-271, and 394-397 in revised manuscript).
Q: Table 2 has multiple grammar error, from misspelling concentration to the use of lowercase for the statistical representation.
R: Thank you very much for your comment. I feel sorry for my mistake, and I have revised those sentences (see Line 401-404 in revised manuscript).
Q: Bibliographic search should be improved since only few recent papers are listed.
R: Thank you very much for your great suggestion, I completely agree with your point, and I have checked some newly published literatures and cited some appropriate literature in this manuscript (see Line 306-308, 314-316, 324-334, 339-345, and 358-370 in revised manuscript).
Q: Please use the proper italics when naming Torreya grandis seed coat.
R: Thank you very much for your comment. I feel sorry about my mistakes, and I have changed this word into proper italics and checked throughout the manuscript (see Line 78, 90, 158, 189, 191, 198, 225, and 227 in revised manuscript).
Q: Correct throughout the text error such as CD3OD, ABTS+, among other.
R: Thank you very much for your comment. I have corrected throughout the text error (see Line114, 124, 132, 234, 235, and 246 in revised manuscript).

Round 2
Reviewer 3 Report
The paper was improved.